

# Impacts of glacier changes on precipitation in the Tibetan Plateau

## Qian Lin [1], Jie Chen[1], Deliang Chen[2]

[1] State Key Laboratory of Water Resources Engineering and Management, Wuhan University, Wuhan, China.

[2] Department of Earth Sciences, University of Gothenburg, Gothenburg, Sweden.

*Correspondence to*: Jie Chen (jiechen@whu.edu.cn)

**Abstract**

The Tibetan Plateau (TP) harbors the largest expanse of glaciers at middle and high latitudes globally. Against the backdrop of ongoing global warming, TP glaciers have experienced widespread retreat and significant mass balance alterations in recent decades, raising questions about their impact on regional climate. In this study, we address this knowledge gap by investigating

the magnitude and spatial extent of precipitation responses to glacier changes across the TP using four distinct Weather Research and Forecasting (WRF) simulations reflecting different glacier and climate conditions. Our findings reveal that, on average, mean precipitation (except for winter) tends to diminish by approximately 0.6% to 2.0% during a cold year and increases by about 0.2% to 2.5% during a warm year over most grid cells influenced by glacier alterations. Additionally, glacier changes lead to a reduction (or augmentation) of summer mean precipitation by an average of 0.6% to 5.2% (1.2% to 10.7%)

over different regions of the TP during the cold (warm) years, accompanied by a notable increase of 0.8% to 19.7% in summer extreme precipitation, irrespective of climate conditions. In general, glacier changes exert a more pronounced impact on summer extreme precipitation events than mean precipitation, with an average increase of 1.7% and 4.6% over the whole TP during the cold and warm years, respectively. Moreover, glacier changes in warmer climate conditions tend to increase summer precipitation amounts in high-altitude areas when the water supply is adequate.

**1. Introduction**

Glaciers constitute vital components of the cryosphere and serve as sensitive indicators of climate change (Yao et al., 2012; Brun et al., 2017; Su et al., 2022b). Simultaneously, they play a pivotal role in regulating the seasonal water cycle, influencing downstream regions by storing and gradually releasing water resources (Immerzeel et al., 2010; Jansson et al., 2003; Marzeion et al., 2014). Over recent decades, global climate warming has instigated the widespread retreat of glaciers, triggering a cascade

of far-reaching consequences. These effects encompass not only alterations in global sea levels (Hugonnet et al., 2021), regional hydrological dynamics (Pritchard, 2019), and the increased potential for natural hazards (Kaab et al., 2018), but also



substantial impacts on human societies (Su et al., 2022a). In addition, some studies have identified several tipping elements in the climate system, and some of these involve changes in the cryosphere (Mckay et al., 2022; Liu et al., 2023). Consequently, the repercussions of glacier changes have assumed global significance and have garnered increasing attention from the

scientific community and beyond.

Often referred to as the "Third Pole" of our planet, the Tibetan Plateau (TP) and its encompassing regions host the most extensive accumulation of snow and glaciers outside of the polar regions (Kang et al., 2010; Yao et al., 2019). These glacier assemblages constitute the very heart of China's and indeed Asia's glacier resources (Yao et al., 2013) and are aptly termed the "Solid Reservoir" of the TP. However, the relentless advance of global warming has unveiled an overarching and accelerating

trend of glacier retreat across the TP since the 1990s (Li et al., 2019; Maurer et al., 2019; Wang et al., 2019; Yao et al., 2012). Moreover, it has been suggested that the snow cover over the TP has been losing stability and approaching a tipping point since 2008 (Liu et al., 2023). This phenomenon exerts direct and profound impacts on the runoff over some Asian major rivers, including the Indus, Brahmaputra, and Yangtze (Immerzeel et al., 2010; Wang et al., 2021a). It is noteworthy that the changes in glacier status exhibit a mosaic of patterns, intricately shaped by regional climates, environmental conditions, and topography

(Li et al., 2019; Garg et al., 2017; Scherler et al., 2011). For instance, the Himalayas and southeastern TP have borne the brunt of glacier shrinkage, experiencing the most pronounced retreat, while the central Karakoram, western Kunlun and eastern Pamir mountains have exhibited the least retreat (Bhattacharya et al., 2021; Shean et al., 2020; Wang et al., 2019; Kaab et al., 2015). For example, the study of Brun *et al*. (2017) shows that Nyainqentanglha mountain (−4.0±1.5 Gt/a) and Kunlun mountain (1.4±0.8 Gt/a) experienced the most and the least glacier mass changes between 2000 and 2016, respectively.

Although most glaciers in the TP have retreated over the past decades, glaciers in the western Kunlun, central Karakoram, and eastern Pamir mountains remained in balance until recently, which is the phenomenon known as "Karakoram anomaly" (Farinotti et al., 2020). However, the data from 2015-2020 suggests that the "Karakoram anomaly" in these areas has almost come to an end, and the glacier mass losses have prevailed in recent years due to the increase in summer temperature over the TP (Bhattacharya et al., 2021; Hugonnet et al., 2021). In general, the magnitude of shrinkage diminishes as one traverses from

the Himalayas and the southeastern TP toward the continental interior (Yao et al., 2012; Treichler et al., 2019).

Furthermore, several studies have undertaken projections regarding future glacier changes in the TP (Miles et al., 2021; Rounce et al., 2023; Zhao et al., 2023; Kraaijenbrink et al., 2017; Zhao et al., 2014). While varying glacier models have produced somewhat differing outcomes, a consistent trend emerges: the TP is set to experience continued reductions in glacier area and mass. For instance, Kraaijenbrink *et al*. (2017) estimated that a 1.5°C global warming scenario would lead to a TP air

temperature increase of 2.1±0.1°C, resulting in a substantial 36±7% glacier mass loss and a corresponding 36±8% reduction in glacier area by the year 2100. Additionally, research by Miles *et al*. (2021) demonstrated that, even in the absence of 21st-



century warming, 21±1% of ice volume will diminish by 2100 due to climatic-geometric imbalances in the TP. These projections underscore the inevitability of rapid glacier reductions in the TP, which will not only reshape the region's land cover but also exert far-reaching influences on the local and regional climate through complex thermodynamic processes.

While numerous studies have examined the responses of TP glaciers to climate change, only a limited number have delved into the intricate feedback mechanisms through which glacier changes influence regional climate (Lin et al., 2021a; Ren et al., 2020; Potter et al., 2018). For instance, Ren *et al*. (2020) conducted two sets of experiments with the resolution of 3 km in the southeastern TP, one considering the presence of glaciers and the other without, utilizing the Weather Research and Forecasting (WRF) Model. Their findings highlighted that the absence of glaciers could lead to an over 20% increase in precipitation in the high-altitude areas of the southeastern TP. Additionally, Lin *et al*. (2021a) identified a negative feedback mechanism by conducting two WRF simulations with the resolution of 2 km in the Himalayas—one with existing glaciers and the other without. This mechanism, rooted in glacier-air interactions, suggests that glacier retreat may result in augmented local precipitation and snow accumulation on the upper reaches of the Himalayan glaciers. However, these studies primarily rely on comparative analyses between two simulations, namely those with and without glaciers, potentially amplifying the climate response to glacier changes, which are gradual processes unfolding over long timescales, spanning decades or even more. As such, there remains a notable gap in comprehensively investigating the impacts of glacier changes that have occurred over the past few decades on regional climate, including precipitation, across the entire TP, particularly under diverse climate conditions.

Therefore, this study systematically quantifies the impacts of glacier changes on both mean daily and extreme summer precipitation patterns across the TP. To achieve this, we conduct a series of Weather Research and Forecasting (WRF) simulations based on paired glacier and climate conditions, resulting in four distinct combinations. By comparing the outcomes of these simulations, we elucidate the intricate relationship between glacier dynamics and precipitation responses. The manuscript is structured as follows: Section 2 provides an overview of the study area, while Section 3 details the data sources and methodology employed in this research. Sections 4 and 5 present the results and discussions, respectively, shedding light on the implications of glacier changes for TP precipitation. Finally, Section 6 offers our conclusions, summarizing the key findings of this study.

## 2. Study Area

The TP, often referred to as the "Water Tower of Asia," serves as the headwater for several vital rivers in the Asian continent, such as the Indus River, the Yangtze River, and the Yellow River, and so on. Shaped by the influences of the South Asian monsoon and intricate topography, the TP exhibits distinct thermal and moisture gradients spanning from southeast to northwest (Sun et al., 2022). Notably, there is a marked increase in mean annual precipitation from the arid northwest (less




than 50 mm) to the lush southeast (exceeding 2000 mm), coupled with a corresponding decrease in mean annual air temperature

from a temperate 20°C in the southeast to a frigid below −6°C in the northwest (Lin et al., 2021b). The rainy season in the TP

predominantly occurs between May and September, accounting for over 80% of the annual precipitation (Curio and Scherer,

2016; Lin et al., 2021b). The study area considered here is confined to the TP within China, delineated within the geographical

coordinates of 73°~105°E and 25°~40°N (Fig. 1), encompassing ten hydrological sub-basins (Zhang et al., 2013).

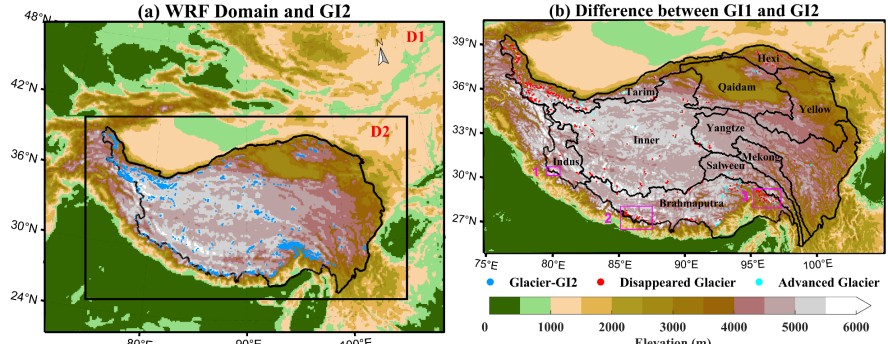

**Figure 1. (a) WRF model domain with glacier distribution of GI2 over the Tibetan Plateau, and (b) Differences between
GI1 and GI2, along with the 10 basins used, and locations of three typical regions selected in this study.**

## 3. Data and Methodology

### 3.1 Data

#### 3.1.1 Glacier Inventory Datasets

To characterize diverse glacier conditions, we utilized datasets from the first Chinese glacier inventory (GI1) and the second

Chinese glacier inventory (GI2) in this study. GI1 and GI2 furnish crucial data, including glacier outlines and fundamental

parameters, enabling us to gain insights into glacier status in China during two distinct time periods.

GI1 was meticulously compiled using topographic maps and aerial photographs spanning the 1960s to the 1980s (Shi et al.,

2009). It represents a cornerstone dataset for understanding the distribution and fundamental characteristics of glaciers in China

during the earlier period (Su et al., 2022b). In the TP within China, GI1 identified 36,793 glaciers covering an area of 49,873.44

km$^2$ and possessing a total volume of 4,561.3857 km$^3$ during the 1960s to 1980s (Pu et al., 2004). These glaciers constitute a

substantial portion, comprising 79.5% in terms of number, 84% in terms of area, and 81.6% in terms of volume of all glaciers

in China (Yao et al., 2013). The GI1 dataset provided by the Key Laboratory of Remote Sensing of Gansu Province, Chinese

Academy of Sciences was subjected to rigorous digitization and quality control processes conducted by the Chinese Glacier



Information System (Li et al., 2008). While it is important to note that the traditional mapping method and digitization process

may introduce minor biases to GI1, the dataset's quality is considered adequate for glacier assessments (Wu and Li, 2011).

GI2 was meticulously compiled using remote sensing images from 2004 to 2011, employing advanced geographic

information system (GIS) techniques (Guo et al., 2015; Liu et al., 2015). It is publicly available through the National Tibetan

Plateau Data Center (http://data.tpdc.ac.cn). GI2 reveals that, during the 2000s to 2010s, there were a total of 48,571 glaciers

covering an aggregate area of 51,800 km$^2$ in China (Liu et al., 2015). In comparison to GI1, this dataset highlights a notable

15% average reduction in the total glacier area across the TP during this period (Yao et al., 2013). Following rigorous validation,

the area error for all compiled glaciers in GI2 was found to be within ±3.2%, with positioning errors of ±10m for clean-ice

glaciers and ±30m for debris-covered glaciers (Guo et al., 2015). Consequently, the GI2 dataset is deemed sufficiently reliable

for the purposes of this study.

### 3.1.2 Precipitation Data for Model Evaluation

Given the limited availability of gauge observations in the TP region, we turned to the Global Precipitation Measurement

mission (GPM) as a reference for assessing the performance of the Weather Research and Forecasting (WRF) model in

simulating precipitation patterns. Launched in 2014 (Hou et al., 2014) through collaboration between the National Aeronautics

and Space Administration (NASA) and the Japanese Aerospace Exploration Agency (JAXA), GPM offers a half-hourly

precipitation product featuring a horizontal resolution of approximately 0.1°, equivalent to approximately 10 kilometers.

Specifically, we utilized the Integrated Multi-satellite Retrievals for the GPM (IMERG) final run product in this study, which

has garnered recognition for its superior accuracy and reliability (Huffman et al., 2020).

### 3.2 Methodology

### 3.2.1 Experimental Design

The WRF model (Powers et al., 2017; Skamarock and Klemp, 2008) is a state-of-the-art atmospheric modeling system

designed to facilitate numerical weather predictions and regional climate simulations. It has enjoyed widespread utilization in

prior studies centered around the TP region (Lin et al., 2021b; Ou et al., 2020; Zhao et al., 2022). For this study, we employed

the WRF model version 4.1.2.

To assess the effects of glacier changes on simulated precipitation, we introduced two distinct glacier conditions, namely

GI1 and GI2, into the WRF simulations. In the process of configuring these glacier conditions within the WRF model, the grid

cells corresponding to glacierized areas, specified with a land use type of 'SNOW/ICE,' were initially reclassified as bare

ground. Subsequently, the land use type for grid cells situated at the locations of TP glaciers in GI1 and GI2 was designated as



'SNOW/ICE.' In total, the TP encompassed 3,024 glacierized grid cells covering an area of 48,384 km² in GI1 and 2,494 glacierized grid cells with a total area of 39,904 km² in GI2 (refer to Fig. 1). Notably, there was a 17.5% disparity in glacier area across the TP between GI1 and GI2 as implemented within the WRF model.

To explore whether the response of precipitation to glacier changes is influenced by climate conditions, we incorporated two distinct climate scenarios into our analysis. These two climate conditions were selected based on the regional mean

temperature data for the TP spanning from 1960 to 2015, derived from the fifth generation of the European Centre for Medium-Range Weather Forecasts (ECMWF) atmospheric reanalysis of the global climate known as ERA5 ((C3s), 2017) (see Fig. S1). Specifically, we designated a "cold year," exemplified by 1967, and a "warm year," exemplified by 2010, as the climate backgrounds when conducting the WRF simulations under both GI1 and GI2 glacier conditions. Consequently, we established four distinct combinations of glacier and climate conditions, namely GI1 in 1967 (1967-GI1), GI1 in 2010 (2010-GI1), GI2 in

1967 (1967-GI2), and GI2 in 2010 (2010-GI2). It's worth noting that all four simulations shared identical WRF configurations, allowing us to interpret disparities in precipitation between two simulations conducted in the same year but under differing glacier conditions as indicative of the potential impacts of TP glacier changes on simulated precipitation.

### 3.2.2 Model Configuration

In the WRF model, a one-way double-nested domain was configured, consisting of grid cells measuring 321×248 (horizontal

resolution of 12 km, referred to as D1) and 775×445 (horizontal resolution of 4 km, referred to as D2) (see Fig. 1). This analysis specifically utilized the hourly output from the inner domain, D2. A total of thirty-eight vertical levels were employed in the vertical direction.

The chosen physical parameterization schemes followed the conclusions of Prein *et al*. (2023), showcasing superior performance over the TP. These schemes encompass the rapid radiative transfer model for general circulation models (RRTMG)

scheme for both longwave and shortwave radiation (Iacono et al., 2008), the Kain-Fritsch cumulus potential scheme (Berg et al., 2013) in D1 (no cumulus parameterization scheme was used in D2), Thompson scheme for microphysics (Thompson et al., 2008), Mellor-Yamada Nakanishi and Niino Level 3 (MYNN3) for the planetary boundary layer (Nakanishi and Niino, 2006), and the Noah land surface model for land surface representation (Chen and Dudhia, 2001).

ERA5 was employed as forcing data for the WRF model, providing hourly temporal resolution and a spatial resolution of

0.25º. Each simulation commenced at 00:00 UTC (Universal Time Coordinated) on December 25 of the preceding year (i.e., 00:00 UTC on December 25, 1966, or 00:00 UTC on December 25, 2009) and concluded at 00:00 UTC on January 1 of the subsequent year (i.e., 00:00 UTC on January 1, 1968, or 00:00 UTC on January 1, 2011). The initial 7 days of each simulation were designated as spin-up periods and were not included in the subsequent analysis.



### 3.2.3 Evaluation Metrics

To assess the accuracy of the WRF model in simulating precipitation, we employed two widely recognized statistical metrics: the Spatial Correlation Coefficient (SCC) and the Mean Relative Error (MRE). The SCC evaluates the model's capability to replicate the spatial distribution of precipitation. It quantifies the degree of agreement between the simulated precipitation patterns and observed data. The MRE, on the other hand, serves to quantify the extent of deviation between the simulated precipitation values and the observations. It is computed by taking the difference between the WRF simulation and the GPM

data (i.e., WRF simulation minus GPM) and then dividing this difference by the precipitation value in the GPM dataset.

Our analysis was conducted on a seasonal basis, encompassing spring (March to May), summer (June to August), fall (September to November), and winter (December to February), thereby providing a comprehensive evaluation of the model's performance across different seasons.

### 4. Results

**4.1 Evaluation of Modeled Precipitation**

In the process of evaluating the WRF model's performance in simulating precipitation over the TP, the 2010-GI2 simulation was considered the closest representation to the current conditions. A comparison was made between the mean daily precipitation of the 2010-GI2 simulation and the corresponding data from the Global Precipitation Measurement mission (GPM) for the year 2010. The spatial distributions of mean daily precipitation from both the GPM and 2010-GI2 simulation are

presented in Fig. 2. Additionally, Figure S2 displays the MRE values, which quantify the relative deviation between the simulated mean daily precipitation and the corresponding GPM observations.

Figure 2 shows that precipitation is primarily concentrated in the southeastern TP across all four seasons, with mean daily precipitation exceeding 10 mm/d in 2010. In comparison with the GPM data, the WRF model generally captures the spatial patterns of mean daily precipitation, with SCC values ranging between 0.39 and 0.72 across all seasons. However, wet biases

were observed in most grid cells for simulations in all four seasons, notably in winter (Fig. S2). Specifically, the mean MRE of the mean daily precipitation in 2010 was calculated to be 20.1%, with the highest mean MRE recorded in winter at 114.3%, and the lowest mean MRE observed in summer at −3.0%. Similar wet biases have been noted in various prior studies (Lin et al., 2018; Wang et al., 2021b; Lin et al., 2021b), likely attributed to the increased uplift of small and medium-sized terrain during the simulation, driven by the complex terrain of the TP (Ren et al., 2020). Nevertheless, these results collectively affirm

that the WRF model reliably replicates the spatial pattern and magnitude of precipitation over the TP.





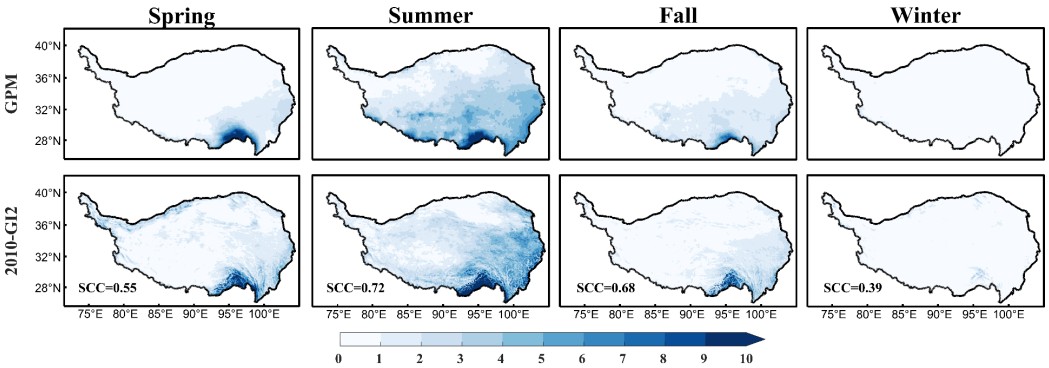

**Figure 2**. **Spatial distributions of mean daily precipitation in four seasons and the whole year of 2010 (warm year): GPM data (top row), 2010-GI2 data (bottom row). Spatial Correlation Coefficient (SCC) is calculated based on the GPM.**

### 4.2 Impacts of Glacier Changes on Precipitation and Thermodynamic Variables

#### 4.2.1 Mean Daily Precipitation

Focusing on daily precipitation, we initiated our analysis by examining the responses to glacier changes. Figure 3 presents the spatial distributions of mean daily precipitation for 1967-GI1 and 2010-GI1, alongside their relative differences under two glacier conditions (i.e., GI1 and GI2) and two climate conditions (i.e., 1967 and 2010). These relative differences were calculated by dividing the mean daily precipitation differences (1967-GI2 minus 1967-GI1 or 2010-GI2 minus 2010-GI1) by the mean daily precipitation of 1967-GI1 or 2010-GI1.

In general, the spatial variabilities of differences in mean daily precipitation between GI1 and GI2 are substantial for both 1967 and 2010, showcasing no discernible spatial distribution patterns. Upon comparing the mean daily precipitation of 1967-GI1 and 2010-GI1 with those of 1967-GI2 and 2010-GI2, a notable finding emerges: the impacts of glacier changes on mean daily precipitation are more pronounced in summer and fall compared to spring and winter. Specifically, the relative differences in mean daily precipitation range between −40% and 40% in summer and fall across most TP regions, contrasting with a narrower range of −10% to 10% observed in spring and winter (excluding the southwest TP). On average, mean daily precipitation experiences a decrease of 1.2% in 1967 and an increase of 1.0% in 2010 across all TP grids. However, it is noteworthy that the relative differences in mean daily precipitation between the two glacier conditions are not statistically significant ($p<0.1$) for almost all grids, implying that the glacier changes between GI1 and GI2 have a minor effect on the magnitude of mean daily precipitation.

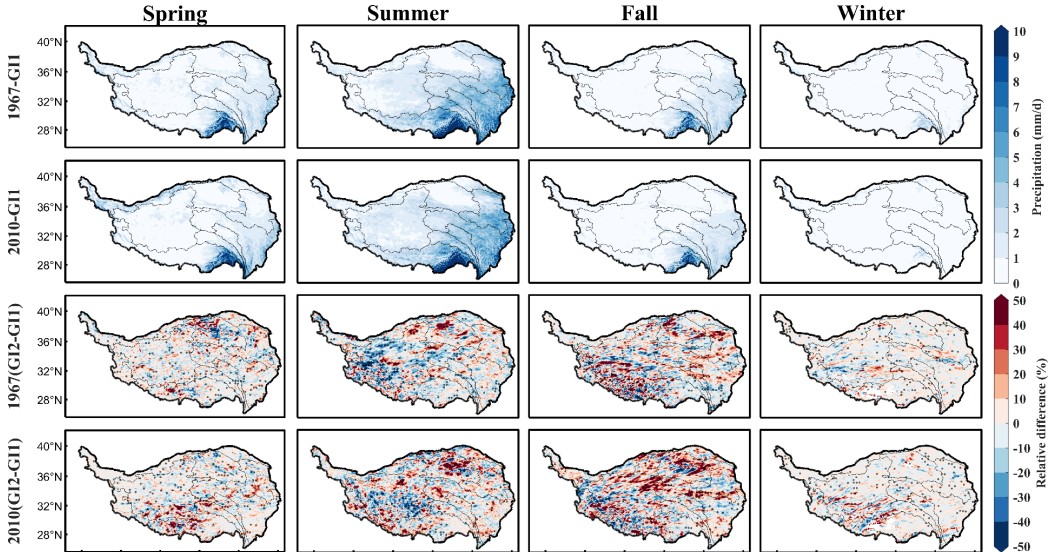

**Figure 3**. **Spatial distributions of mean daily precipitation for 1967-GI1 (top row), 2010-GI1 (second row), and relative differences between GI1 and GI2 in 1967 (third row) and 2010 (bottom row). The stippled regions indicate statistically significant differences at the 10% significance level.**

### 4.2.2 Summer Extreme Precipitation

Given the significance of summer precipitation, we delved deeper into the impacts of glacier changes on summer extreme precipitation. Extreme precipitation was defined as the 95th percentile of daily precipitation during the summer season. In Fig. 4, we present the spatial distribution of summer extreme precipitation for 1967-GI1 and 2010-GI1, along with the relative differences between GI1 and GI2 for the years 1967 and 2010. These relative differences were computed by dividing the summer extreme precipitation differences (1967-GI2 minus 1967-GI1 or 2010-GI2 minus 2010-GI1) by the summer extreme precipitation of 1967-GI1 or 2010-GI1.

The results illustrate a decline in summer extreme precipitation from the southeast to the northwest of the TP in both 1967 and 2010, mirroring the spatial distribution observed for mean daily precipitation. The highest summer extreme precipitation is observed to be over 30 mm/d in the southeastern TP, while it remains under 3 mm/d in the northwestern TP. The relative differences in summer extreme precipitation between the two glacier conditions exhibit significant spatial variability, ranging from −50% to 50%. As a result of glacier changes, summer extreme precipitation experiences an average increase of 1.7% and 4.6% over the entire TP in 1967 and 2010, respectively.

Notably, the relative difference in extreme precipitation is generally larger than that of mean daily precipitation in summer, although they share a similar spatial distribution pattern in terms of relative difference. These findings emphasize that glacier



changes exert a more substantial influence on extreme precipitation compared to mean daily precipitation during the summer season.

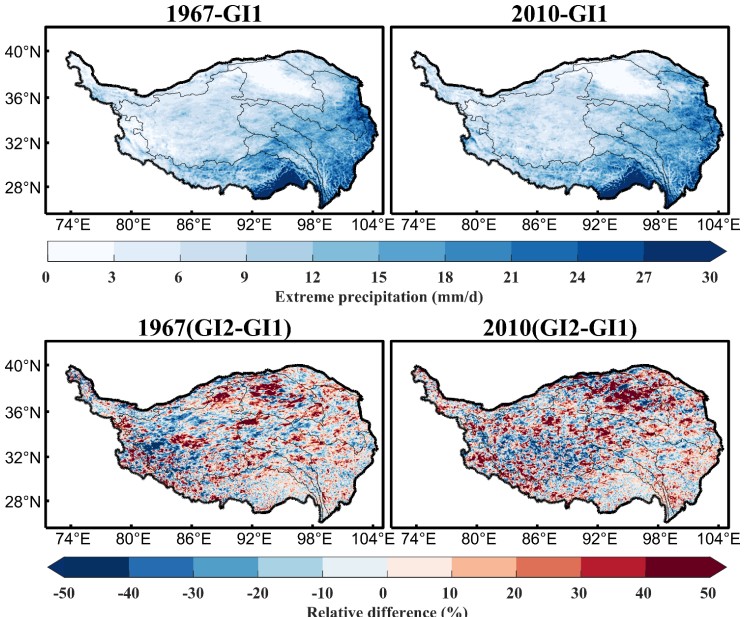

**Figure 4. Spatial distributions of summer extreme precipitation for 1967-GI1, 2010-GI1, and relative differences between GI1 and GI2 in 1967 (cold year) and 2010 (warm year).**

### 4.2.3 Thermodynamic Variables

Glacier changes represent a form of land cover alteration significantly affecting regional climate by modifying thermodynamic processes (Horton et al., 2015; Lin et al., 2021c). Hence, we also examined key thermodynamic variables. Figures S3-S8 depict the spatial distributions of mean daily air temperature at 2 m (T2), net radiation (NR), sensible heat flux (SH), latent heat flux (LH), convective available potential energy (CAPE), and convective inhibition (CIN) for 1967-GI1 and 2010-GI1 in four seasons across the TP, alongside their differences between GI1 and GI2 for the years 1967 and 2010.

In Fig. S3, the differences in mean daily T2 between GI1 and GI2 range from −0.1°C to 0.1°C in spring and winter for most TP regions. Conversely, the differences are much larger in summer and fall, spanning between −0.5°C and 0.5°C. Additionally, the mean daily T2 differences reach the significance level ($p<0.1$) for spring, fall and winter, but not for summer. The spatial distributions of other thermodynamic variables are similar to T2, showcasing a decline from the southeast to the northwest of the TP. Furthermore, the differences in these thermodynamic variables tend to be more pronounced in summer and fall (Figs. S4-S6). During these two seasons, the differences in NR, SH, and LH between GI1 and GI2 hover around ±8W/m², ±8W/m², and ±4W/m² in most TP areas, approximately doubling those in spring and winter. Moreover, the differences in mean daily



NR, SH, and LH also reach significance ($p<0.1$) for most TP regions.

Concerning summer mean daily CAPE and CIN, their spatial distributions correspond to the pattern of summer extreme precipitation (Figs. S7-S8). Regions with more extreme precipitation exhibit higher CAPE and lower CIN, and vice versa. The differences in summer mean daily CAPE and CIN induced by glacier changes are statistically significant ($p<0.1$) for only a few grid cells in both 1967 and 2010. Notably, the Inner and Qaidam basins show substantial differences in summer mean daily CAPE and CIN, aligning with the regions where extreme precipitation differences are notable.

Figure 5 displays the differences in precipitation and key thermodynamic variables (T2, NR, SH, LH, CAPE, and CIN) triggered by glacier changes across the ten basins during summer. In 1967, glacier changes lead to a reduction in summer mean daily precipitation by 0.6%~5.2% in six out of ten basins (excluding the Indus, Qaidam, Salween, and Yellow basins). However, in 2010, seven out of ten basins (excluding the Inner, Salween, and Tarim basins) experience an increase of 1.2%~10.7% in summer mean daily precipitation. Additionally, summer extreme precipitation increases by 0.8%~10.9% in seven out of ten

basins (excluding the Inner, Mekong, and Tarim basins) in 1967, and by 0.8%~19.7% for all TP basins in 2010. When comparing the results between 1967 and 2010, it becomes evident that the impacts of glacier changes on precipitation are not consistent under these two climate conditions. Despite glaciers showing a consistent state of loss across all basins, these results highlight that glacier loss tends to induce more mean and extreme precipitation during summer for most TP basins under warmer climate conditions.

When examining the variations in thermodynamic variables attributed to glacier changes, it becomes evident that the summer mean daily T2 increases in most basins are consistently accompanied by increases in NR, SH, and LH, or vice versa. Mean daily T2 tends to rise by 0℃~0.07℃ for most basins in the cold year, while the opposite trend is observed in the warm year. This behavior may be linked to the LH increase in most basins during warm years, indicating higher evaporation and leading to a decrease in T2. In the cold year, some basins (such as the Brahmaputra, Mekong, and Salween basins) exhibit an

increase in T2, NR, SH, and LH but a decrease in mean daily precipitation. This pattern may be because the increase in SH is insufficient to promote cloud formation, and the amount of water vapor is also not enough to support the production of more precipitation.

Additionally, summer extreme precipitation increases alongside the decrease in both CAPE and CIN in certain basins (such as the Indus basin in the cold year and the Brahmaputra, Salween, Yangtze, and Yellow basins in the warm year). While the

values of CIN are relatively small over the TP, this phenomenon may be attributed to the fact that the decrease in CIN outweighs the decrease in CAPE in these basins, leading to more intense convective activities and resulting in increased extreme precipitation.

In general, summer mean and extreme precipitation are more likely to increase in the warm year compared to the cold year

for most basins. This difference may be attributed to the higher moisture availability and more intense heat exchange between

the atmosphere and the surface during warm years. Additionally, glacier-induced increases in LH may provide additional

moisture to the air in most basins during warm years, further contributing to increased precipitation.

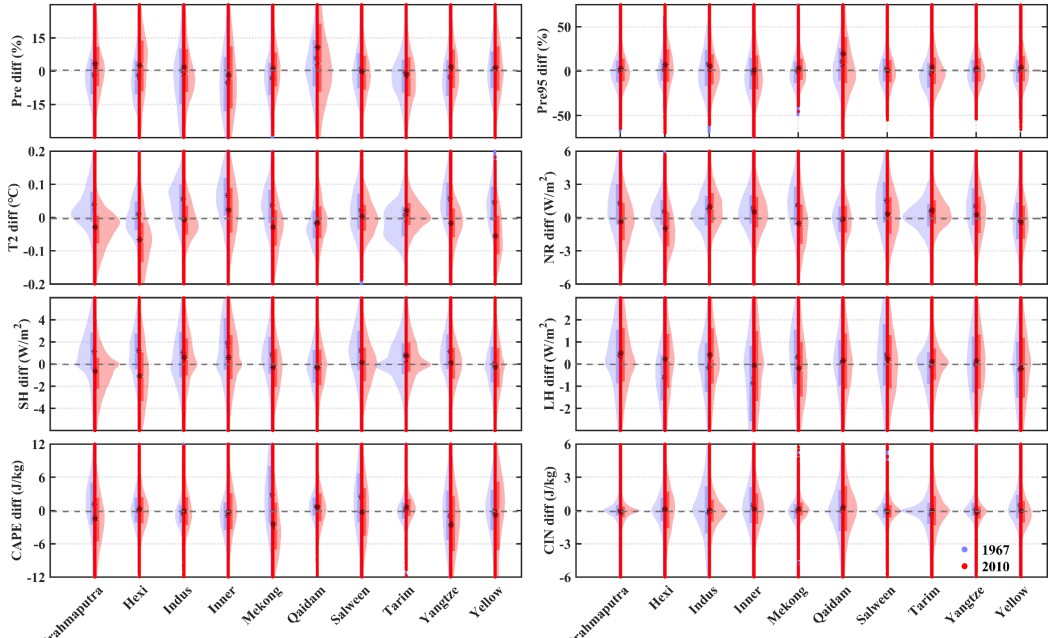

**Figure 5. Violin plot of mean daily precipitation (Pre), extreme precipitation (Pre95), air temperature at 2 m (T2), net radiation (NR), sensible heat flux (SH), and latent heat flux (LH), convective available potential energy (CAPE), and**

**convective inhibition (CIN) differences between GI1 and GI2 in summer of 1967 (cold year) and 2010 (warm year) over the 10 basins of the Tibetan Plateau.**

### 4.2.4 Local Precipitation and Thermodynamic Variables

To delve into the localized climatic impacts of glaciers, we analyzed the differences in mean precipitation and key thermodynamic variables between the two glacier conditions over the grid cells where glaciers either advanced (84 grid cells)

or disappeared (614 grid cells). Figure 6 illustrates these differences for mean daily precipitation, T2, NR, SH, and LH between the two glacier conditions in 1967 and 2010 over these specific grid cells.

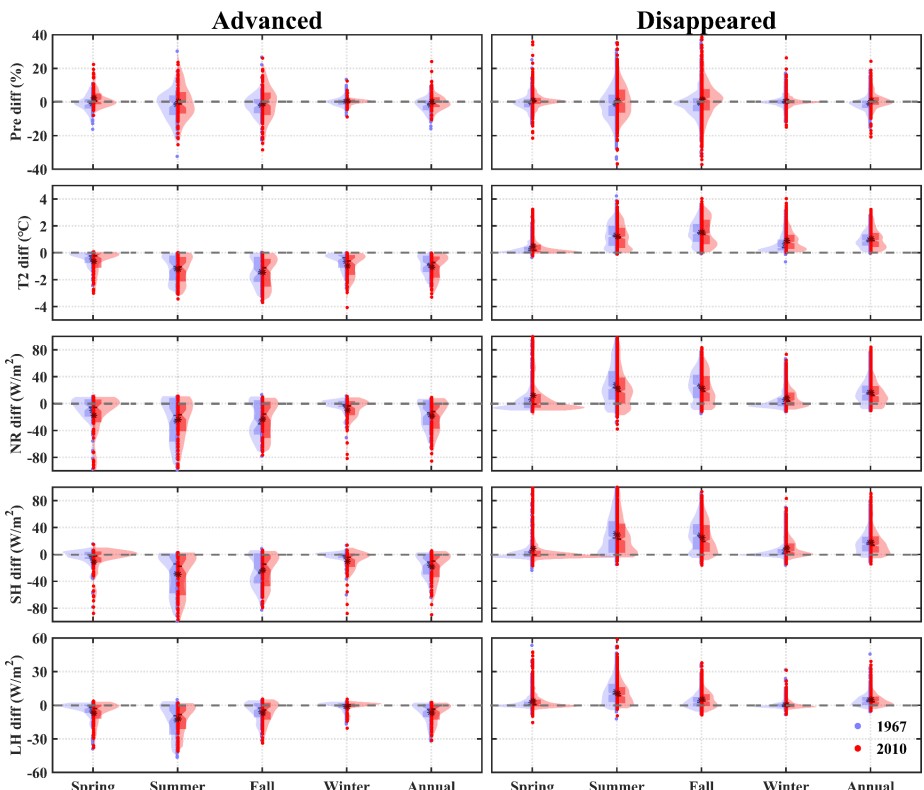

**Figure 6. Violin Plot of mean daily precipitation (Pre), air temperature at 2 m (T2), net radiation (NR), sensible heat flux (SH), and latent heat flux (LH) differences between GI1 and GI2 in four Seasons of 1967 (cold year) and 2010 (warm year) for glacier-advanced and glacier-disappeared grid cells.**


The analysis reveals that in most glacier-advanced grid cells across all seasons in both 1967 and 2010, there is a decrease in mean daily T2, NR, SH, and LH, while the opposite holds true for glacier-disappeared grid cells. These altered thermodynamic variables at the local scale result in a decrease (increase) in mean daily T2 by an average of 0.5℃~1.5℃ (0.3℃~1.5℃) primarily due to glacier effects. Moreover, the mean daily T2 differences induced by glacier changes are generally more

pronounced in 2010 than in 1967, suggesting that the influence of glaciers on mean daily T2 might be amplified under warmer climate conditions. Additionally, mean daily precipitation tends to decrease by an average of 0.6%~2.0% in both glacier-advanced and glacier-disappeared grid cells in 1967 (except for winter). However, in 2010, the results diverge, with mean daily precipitation increasing by an average of 0.2%~2.5% for most glacier-changed grid cells (except for glacier-advanced grid cells in fall). As elucidated in Section 4.2.3, when all these thermodynamic variables increase, the likelihood of cloud formation

may rise, and heat exchanges between the surface and atmosphere are intensified, potentially leading to increased precipitation in most cases. In the cold year, the reduced precipitation in the glacier-disappeared grid cells may stem from the increased



thermodynamic variables not being sufficient to create conditions for heightened precipitation, primarily utilized to elevate mean daily T2. Conversely, the increased precipitation in glacier-advanced grid cells in a warm year might be attributed to interactions between adjacent grid cells. Specifically, the number of glacier-advanced grid cells is substantially fewer than that

of glacier-disappeared grid cells, and most glacier-advanced grid cells are proximate to glacier-disappeared grid cells, causing mutual influence between adjacent grid cells. Overall, these results underscore that glacier loss may lead to heightened local-scale precipitation under warmer climate conditions.

**4.3 Impacts of Glacier Changes on Precipitation for Typical Regions**

Based on above analyses, we found that glacier changes have larger impacts on precipitation under warmer climate

conditions, especially in summer. Considering the fact that the analysis of simulations in warmer conditions may have more practical significance for future climate change, two additional simulations are generated for the summer of the warm year (i.e., 2010) by perturbing the initial conditions of the WRF model. To be specific, these two sets of simulations shared identical WRF configurations as the previous simulation. Based on the average of the three sets of simulations for the summer of the warm year (i.e., 2010), the impacts of glacier changes on precipitation for three typical regions are further analyzed. The three

typical regions (Fig. 1) that experienced relatively large glacier shrinkage are selected. In addition, the scales of glacier change are mostly larger than the simulated resolution (i.e., 4 km) for these three regions. Specifically, the relative differences in glacier area between GI1 and GI2 are −18.5%, −28.7%, and −12.4% in these three regions, respectively.

Figures 7-9 present the spatial distributions of summer precipitation amounts and water vapor flux (WVF) for 2010-GI1, alongside their differences between two glacier conditions (i.e., GI1 and GI2) in 2010 for three typical regions, respectively.

The WVF is the major water vapor source for summer precipitation over the TP (Zhang et al., 2017). In Fig. 7, it can be seen that glacier changes lead to an increase in summer precipitation amounts for the south and the east of Region 1, and a decrease for other areas. Specifically, the meridional average of the maximum relative difference in summer precipitation amounts caused by glacier changes can be up to 12.8% in Region 1. Precipitation in Region 1 is dominated by the westerlies, which carry the air masses from the west to the east. At high altitudes, the WVF is smaller than that at low altitudes with the value of

$10 \, \text{kg} \, \text{m}^{-1}\text{s}^{-1}$. This is because the increased terrain height hinders the WVF transport eastward. Glacier changes tend to decrease the summer WVF in Region 1, but only to a slight degree, with values smaller than $1 \, \text{kg} \, \text{m}^{-1}\text{s}^{-1}$ for most areas. As shown in Fig. 7f, the relatively strong differences in WVF mainly occur at a height of approximately 500 hPa, dominating the pattern of summer WVF difference in Region 1. Moreover, the WVF is mainly concentrated in the lower and middle troposphere (below 400 hPa) (Liu et al., 2019), which is highly correlated with precipitation. Thus, combining the differences in summer

precipitation amounts and WVF due to glacier changes, we found that the reduced WVF in the lower and middle troposphere



corresponds to the reduced precipitation amounts, and vice versa. Compared to Region 1, Regions 2 and 3 have a more abundant water supply from the Indian summer monsoon, showing more summer precipitation amounts, with maximum values up to 800 mm and 1000 mm, respectively. Affected by significant terrain fluctuations, the summer WVF transport is hindered, resulting in lower WVF in most areas of Region 2. The summer WVF is higher in Region 3, with values of more than 30 kg

$m^{-1}s^{-1}$ for most areas. Additionally, glacier changes tend to increase in summer precipitation amounts by 0~30% for Regions 2 and 3, which are accompanied by the increase of summer WVF. The patterns of summer WVF difference between GI1 and GI2 in Regions 2 and 3 are also associated with the WVF difference at a height of approximately 500 hPa. Same as Region 1, the differences in summer precipitation amounts in Regions 2-3 are closely related to the WVF difference in the lower and middle troposphere.

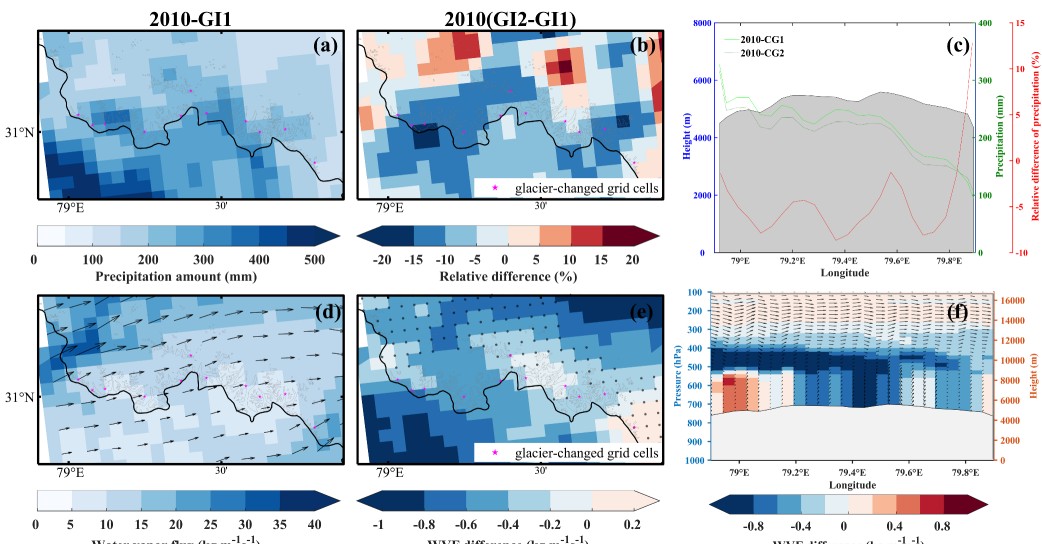


**Figure 7. Spatial distributions of (a) summer precipitation amounts and (d) total column water vapor flux for 2010-GI1, and (b) relative difference in precipitation amounts and (e) the difference in total column water vapor flux between GI1 and GI2 in 2010 for Region 1. The gray curve represents the glacier boundary. The stippled regions indicate statistically significant differences at the 10% significance level. The (c) meridional mean summer precipitation**

**amounts and its relative difference between GI1 and GI2 in 2010 for Region 1. The (f) meridional vertical profile of summer water vapor flux difference between GI1 and GI2 in 2010 for Region 1. The arrow represents meridional-vertical or zonal-vertical wind.**



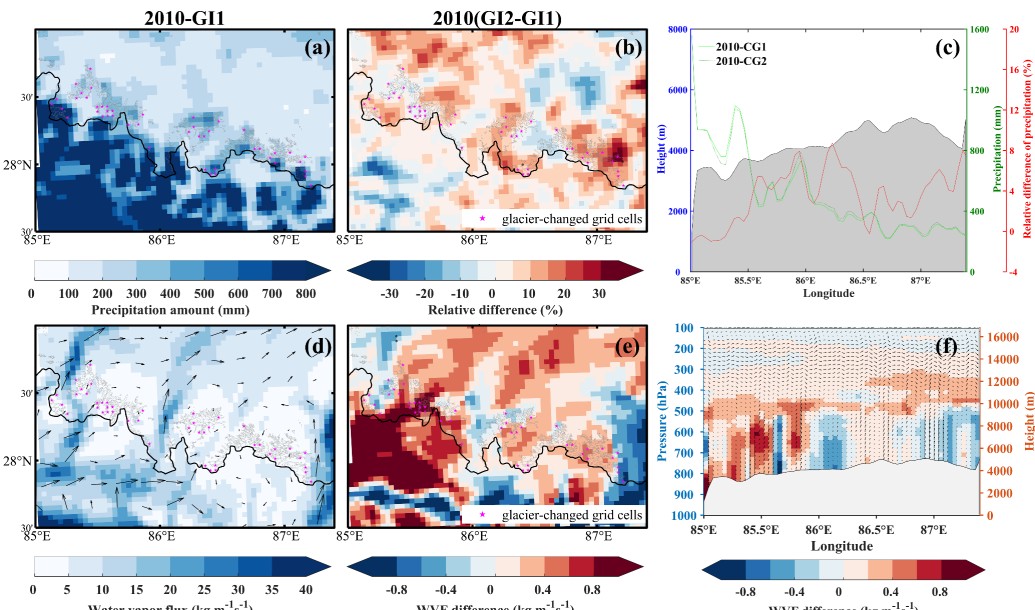

**Figure 8. Same as Figure7, but for Region 2.**

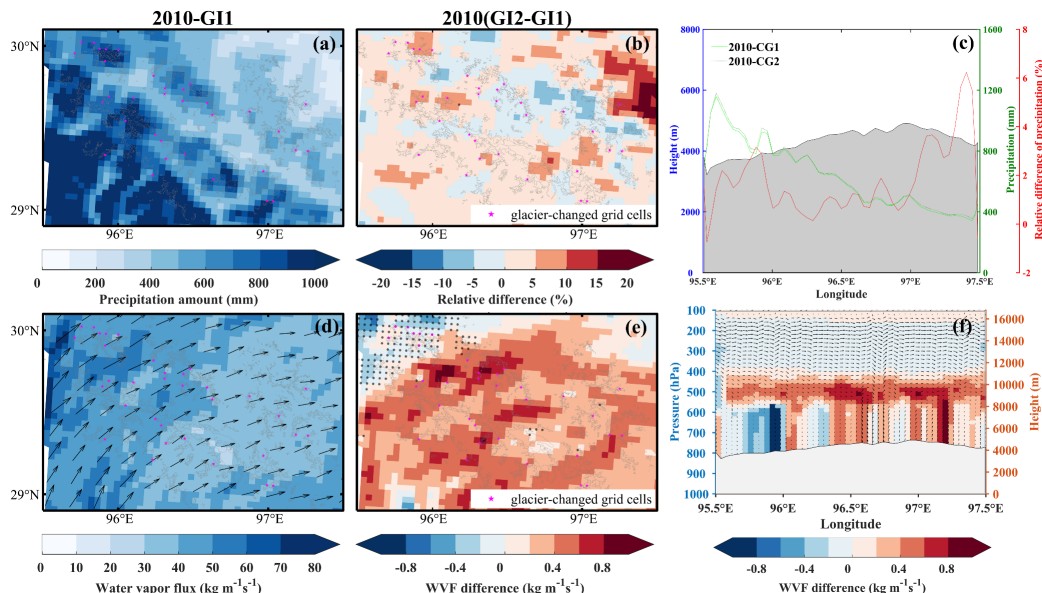


**Figure 9. Same as Figure7, but for Region 3.**

Comparing the results for the three typical regions, we found, glacier loss leads to an overall decrease in summer precipitation amounts under a warmer condition only in Region 1, which are opposite in Regions 2 and 3. This is because Region 1 features higher altitude, lower temperature, and smaller WVF, which are more unfavorable for precipitation.

Meanwhile, glacier loss further reduces the summer WVF, leading to decreased summer precipitation amounts. In addition,



the results of Regions 2 and 3 imply that glacier loss in warmer conditions is likely to increase summer precipitation amounts in high-altitude areas where most glaciers are located and the water supply is adequate. Although the relative differences in summer precipitation amounts are not statistically significant ($p<0.1$) for most grids in Regions 2-3, these results confirm the mechanisms of glacier-air interactions found by others in the Himalayas (Lin et al., 2021) and the southeastern TP (Ren et al.,

2020), which indicate that glacier loss can lead to an increase in precipitation in the local or high-altitude areas.

## 5. Discussion

The investigation of glacier change impacts across two glacier conditions (GI1 and GI2) on simulated precipitation across the TP with differing climate backgrounds has yielded critical insights. The findings indicate that glacier loss leads to heightened mean daily precipitation and summer extreme precipitation in warmer climate conditions across most TP regions.

Moreover, glacier loss has a direct impact on raising local-scale temperatures (T2) by absorbing more solar energy, potentially accelerating the melting of adjacent glaciers. Upon glacier disappearance, warmer air above the former glacier area tends to ascend, generating a local circulation counter to the previously descending glacial wind. This has implications for local climate adjustments through modifications to atmospheric circulations (Ren et al., 2020). Within the context of global warming, this positive feedback loop involving glaciers might expedite localized climate warming and further amplify the repercussions of

glacier loss. Anticipated consequences encompass heightened summer precipitation and meltwater due to glacier loss, likely leading to increased runoff in downstream basins for a certain period that before the "peak meltwater" (Huss and Hock, 2018), thus affecting water resource regulation.

While the impacts of glacier changes on precipitation show no obvious spatial distribution patterns over the TP, the relative differences of precipitation are still very large in some grid cells in both warm and cold years. This indicates that glacier

changes over the TP during the past decades have great influence on precipitation at the local scale, and the glacier-air interactions should not be completely ignored. This is especially true for long-term glacier projections, as it has been found that completely ignoring the glacier-air interactions may overestimate the retreat of glaciers in the TP (Lin et al., 2021a). Therefore, considering the glacier-air interactions may be necessary for conducting a more accurate long-term glacier projection under the background of global warming. However, the mechanisms of glacier-air interactions have not yet been

fully revealed. As the results of three typical regions shown that affected by the terrain, climate conditions, and other factors, the climatic effects of different glaciers may be various. Thus, more research is needed to investigate the mechanisms of interactions between glaciers and the atmosphere.

Results indicate that the Brahmaputra basin have experienced relative more intensive glacier changes, but the precipitation differences caused by glacier changes are not larger than in other basins. Glacier changes influence precipitation primarily





through altering thermodynamic processes, influenced by multiple factors such as topography and water vapor transport (Horton et al., 2015; Lin et al., 2021c). Ren *et al.* (2020) found that glacier loss can strengthen the transportation of warm, moist air to highlands, leading to an increase of more than 20% in precipitation in the high-altitude area of the southeastern TP. However, this may not hold true for all cases. Given that glaciers cover only a small portion of the TP, the effects of glacier changes are limited to their immediate surroundings. Therefore, the regional mean daily precipitation differences are relatively

small in most basins.

Like many modeling studies, this study also has limitations. The resolution of 4 km for simulations is still too coarse for some small-sized glaciers. Consequently, the total areas of glaciers presented in the WRF model are slightly smaller than their actual areas of GI1 and GI2, due to the fact that the size of some glaciers is too small to be represented in the model. However, the glacier area error caused by these small-sized glaciers is very small, and the difference in glacier area between the two

glacier conditions is very close to the actual situation. Therefore, the results can still reflect the climate effects of glacier changes in recent years. On the other hand, the model evaluation shows that wet biases exist in WRF-simulated precipitation, which may be related to the existing limitations of the WRF model in complex terrain (Lin et al., 2021b; Ren et al., 2020). Meanwhile, the 4 km resolution may still not be sufficient to realistically represent the complex terrain and heterogeneity of land use/cover types in the TP (Lin et al., 2018; Lin et al., 2021a). Nonetheless, the bias of modeling is espected to have have

little impacts on conclusions drawn in this study, since the WRF model is driven using the same schemes and set-ups with only differences in glacier coverage in the same climate condition. Thus, the differences in precipitation and thermodynamic variables between the two glacier conditions in the same climate condition can be considered as the impacts of glacier changes.

## 6. Conclusions

This study investigated the impacts of glacier changes on precipitation over the TP. To achieve this goal, four simulations

with different glacier and climate conditions (i.e., 1967-GI1, 2010-GI1, 1967-GI2, and 2010-GI2) were conducted and compared. The following conclusions can be drawn.

(1) Glacier changes have more impact on extreme precipitation than on mean daily precipitation in summer, resulting in an average increase of 1.7% and 4.6% in extreme precipitation in cold and warm years over the TP, respectively.

(2) The climatic effects of glaciers on precipitation is different for two climate conditions, especially in summer. Specifically,

glacier changes lead to differences in mean precipitation and summer extreme precipitation in all basins, ranging between −5.2% and 5.8% (cold year) and between −2.4% and 10.9% (warm year). Additionally, glacier changes in warmer climate conditions tend to increase summer precipitation amounts in high-altitude areas when the water supply is adequate.

(3) The mean daily T2, NR, SH, and LH decrease over most glacier-advanced grid cells in all seasons, while the opposite is

observed in glacier-disappeared grid cells. The combined effects of these thermodynamic changes result in a mean precipitation

decrease by an average of 0.6%~2.0% in both glacier-advanced and glacier-disappeared grid cells in the cold year (except for

winter). Conversely, an average increase of 0.2%~2.5% in mean precipitation is observed for most glacier-changed grid cells

(except for glacier-advanced grid cells in fall) in the warm year.

**Data availability**

The TP boundary data and the second Chinese glacier inventory dataset are available at the National Tibetan Plateau Data

Center (http://data.tpdc.ac.cn). The GPM data is available at the NASA Goddard Earth Sciences Data and Information Services

Center (https://pmm.nasa.gov/data-access/downloads/gpm). The ERA5 data can be accessed from the European Center for

Medium-Range Weather Forecasts (ECMWF) (https://cds.climate.copernicus.eu/).

**Author contributions**

QL contributed to the drafting of this manuscript and the development and implementation of the methodology. JC and DC

contributed to the design and overall framing of the study and revised the draft.

**Competing interests**

The contact author has declared that none of the authors has any competing interests.

**Acknowledgements**

This work was partially supported by the National Natural Science Foundation of China (Grant No. 52079093), the Hubei

Provincial Natural Science Foundation of China (Grant No. 2020CFA100), and the Overseas Expertise Introduction Project

for Discipline Innovation (111 Project) funded by the Ministry of Education and State Administration of Foreign Experts

Affairs P.R. China (Grant B18037). We would like to thank the Key Laboratory of Remote Sensing of Gansu Province, Chinese

Academy of Sciences for providing the first Chinese glacier inventory dataset. The numerical calculations in this paper have

been done on the supercomputing system in the Supercomputing Center of Wuhan University.

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
