# Peer review of "Impacts of glacier changes on precipitation in the Tibetan Plateau"

_EGUsphere, 2024_

## Author Comment (AC1)

Reviewer #1:

In the submitted manuscript, the authors seek to understand how glacier changes in the Tibetan Plateau may have affected precipitation amounts. WRF simulations at a 4km resolution are used, nested within a 12km outer domain which extends well beyond the Tibetan Plateau. Glacier inventories using data collected from the 1960's to the 1980's are compared with a glacier inventory derived from data collected between 2004-2011 as a proxy for glacier change. The authors then compare precipitation amounts under a warm year or a cold year for the two glacier scenarios. The study promises an interesting compliment to existing studies of glacier-precipitation feedback over the Tibetan plateau, considering marginal changes to glacier extent instead of absolute retreat, as studied by Ren et al., 2020 and Lin et al., 2021.

**Reply:** Thank you very much for reviewing our paper and providing insightful comments. Your feedback is both valuable and helpful. We have carefully addressed these comments in our revisions.

When analyzing precipitation, the authors report relative difference in precipitation between the two glacier scenarios. This exaggerates the effect of changing the glacial land surface data, where relative differences for areas receiving little precipitation are sensitive to small changes in absolute precipitation amount. The maps of summer extreme precipitation suffer from the same problem of reporting relative differences though, making it difficult to tell how much changes to glacier extents have impacted precipitation amounts. The authors indeed note that "the relative differences in mean daily precipitation between the two glacier conditions are not statistically significant ($p < 0.1$) for almost all grids". Section 4.3 is then devoted to explaining these differences. This contradicts the previous finding of the daily-mean (or seasonal total) differences not being significant, and the proposed mechanism of WVF is not supported by the plots shown. Furthermore, the authors focus on areas along the crest of the range, while their methodology only altered glacier extents for Chinese glaciers. This methodological quirk rules out the mechanism proposed by Lin et al., 2021 and Ren et al., 2020, which involved the northerly katabatic flows from glaciers to the south of the crest. Because these glaciers remain unchanged in the study, the mechanism proposed by these two earlier studies cannot be responsible for changes to precipitation. The authors still falsely conclude that their results support the findings of these two prior studies. The manuscript thus shows that changing the land surface type for some cells in the domain results in slight changes to precipitation due to a mechanism which has not been rigorously demonstrated.

**Reply:** Considering there are great seasonal and regional variations of precipitation over the Tibetan Plateau, we used the relative difference of precipitation between the two glacier conditions to facilitate comparisons for differences across regions and seasons. Having said these, we do agree with you that absolute values are useful too. Thus, we will add the results of absolute changes in precipitation induced by glacier changes in the revised manuscript.

The works of Lin et al. (2021) and Ren et al. (2020) have a totally different design with

regard to model resolution and domain. The aim is also different. Thus, their results are not directly comparable with this study and we realize now that our previous discussions were not entirely appropriate. As a result, we will not directly compare the results of this study with theirs in the revision. Your comments are much appreciated.

For these reasons I recommend rejection of the manuscript in its current form. I believe that the study does robustly show that recent glacial retreat/advance in the Tibetan Plateau has resulted in local changes to the surface energy balance. These local changes may cause localized changes to convection. In a follow up, the authors could focus on demonstrating the mechanisms through which glacial retreat and advance affects convective storms, and quantify this effect. For example, section 4.2.3 references non-local changes to precipitation fields as a result of local changes to glacier extent and thus the surface energy balance. Illustrating particular convective cells and extreme precipitation events which occur as a result of recently retreated glaciers would be interesting to see. The dependency on grid resolution, as it impacts convective processes and the representation of glacier retreat, could thus also be investigated. This, in combination with a clearer reporting of changes to precipitation amounts, would improve the manuscript.

**Reply:** We see your point. By following your suggestions above, we are confident to deliver an improved manuscript. However, we don't think it is a good strategy to focus on particular convective cells and extreme precipitation events that occur as a result of recently retreated glaciers as these most likely contain a large amount of noise. By looking at mean conditions over a relatively long period (one season), instead of individual extreme events (a few days), we hope to be able to identify robust signals of changes induced by the changed glaciers. Besides, your suggestion to focus on particular convective cells and extreme precipitation events, as well as the sensitivity of the model simulation to model resolution deviate from our aims, although these are interesting topics. Implementing these ideas would require another study.

---

## Author Comment (AC2)

Reviewer #2:
This paper investigates the feedback from glacier changes on precipitation in the Tibetan Plateau (TP) by using four simulations with the Weather Research and Forecasting (WRF) model. The four different simulations represent a "cold" and "warm" year based on regional mean temperature data from 1960 to 2015, with glacier cover data from two periods (1960s-1980s and 2004-2011), respectively. Specifically, the authors model glacier changes by altering the land surface type in WRF to bare ground based on changes in the glacier inventory data. They find that the highest impact of glacier changes can be seen in extreme precipitation events, specifically in the summer, and that the mean precipitation decreases in a cold year but increases in a warm year.

The manuscript has a valuable objective in discussing the impacts of glacier changes on precipitation patterns in this region. However, in its current form, I believe the manuscript has some shortcomings with regard to its methodological setup and the provided explanations in the text. Comments and suggestions are given in the list below.
**Reply:** We would like to thank the reviewer for the constructive and insightful comments. All comments will be incorporated into the revised and the point-by-point responses are presented as follows:

Major comments:

Evaluation of the modeled precipitation (Section 4.1): The authors are presenting an evaluation between the WRF-modeled precipitation and GPM. However, to assess the WRF model performance, I believe it is crucial to first evaluate the modeled precipitation in ERA5 (serving as initial and boundary conditions for WRF). Subsequently, the authors can evaluate whether downscaling with WRF improves the representation in comparison to ERA5 or not. Could you please add a comparison between ERA5 and GPM and then evaluate how much the WRF model improves the simulation of precipitation (e.g., NRMSE or MRE per basin) for a "cold" and "warm" year.
**Reply:** ERA5 is considered one of the best reanalysis products for this region (e.g., Dan et al., 2021; Zhou et al., 2021; Minola et al., 2024), and the added value of WRF in simulating precipitation in the region has been demonstrated by many previous studies (Zhou et al., 2021; Ma et al, 2023). Thus, we did not want to repeat this (important and useful) exercise in this study. Having said these, in the revised version, we will state these clearly with necessary references.

References:
Dan, J., Gao, Y. and Zhang, M. (2021) Detecting and Attributing Evapotranspiration Deviations Using Dynamical Downscaling and Convection-Permitting Modeling over the Tibetan Plateau. Water 13(15), 2096.
Minola, L., Zhang, G.F., Ou, T.H., Kukulies, J., Curio, J., Guijarro, J.A., Deng, K.Q., Azorin-Molina, C., Shen, C., Pezzoli, A. and Chen, D.L. (2024) Climatology of near-surface wind speed from observational, reanalysis and high-resolution regional climate

model data over the Tibetan Plateau. Climate Dynamics 62(2), 933-953.

Ma, M.N., Ou, T.H., Liu, D.Q., Wang, S.Y., Fang, J. and Tang, J.P. (2023) Summer regional climate simulations over Tibetan Plateau: from gray zone to convection permitting scale. Climate Dynamics 60(1-2), 301-322.

Zhou, X., Yang, K., Ouyang, L., Wang, Y., Jiang, Y.Z., Li, X., Chen, D.L. and Prein, A. (2021) Added value of kilometer-scale modeling over the third pole region: a CORDEX-CPTP pilot study. Climate Dynamics 57(7-8), 1673-1687.

Evaluation of absolute precipitation amounts: The authors report relative precipitation changes throughout the manuscript, but don't focus on absolute amounts. This becomes especially important as the authors state that for most grid cells, the results have been statistically insignificant. Can you please specify and discuss absolute amounts?

**Reply:** We see the value of adding absolute values, which was also pointed out by reviewer #1. We will add the absolute amounts of precipitation changes in the revised manuscript.

Modifications of SNOW/ICE grid cells in WRF: The authors mention that they modified grid cells in WRF (SNOW/ICE or bare ground) in order to model glacier changes in Gl1 and Gl2 (Section 3.2.1). However, from the methodological description it is unclear how this modification has been performed. Have you modified the grid cells in both D1 and D2 (i.e., for D1 within the intersection of D1 with D2) or only D2? Page 6 lines 135-136 refer to D2 (I assume). I believe it is crucial to also change the grid cells in the parent domain (D1).

**Reply:** Sorry for the confusion. In fact, we modified the grid cell in both D1 and D2 in the original manuscript. Specifically, before running the WRF model, the grid cells were modified. After that, we run geogrid.exe, which defines model domains and interpolates static geographical data to the grids. We will clarify these details mentioned above in the revised manuscript.

Moreover, the authors only change glacierized grid cells confined to the TP within China, but they do not change grid cells in WRF in other areas (e.g., in all of D1), which is a physically unrealistic setting. Can the authors please explain in detail their reasoning and possible implications of this choice of model setup on the results?

**Reply:** We only changed glacierized grid cells confined to the TP within China, as the glacier dataset only covers China.

Selection of WRF physics parameterization schemes and albedo representation (Section 3.2.2): The authors mention that they follow the physics parameterizations of Prein et al. (2023). Please provide an overview of their findings and sensitivity tests for the simulation of precipitation and discuss their results for different seasons in order to justify selecting this specific WRF configuration.

**Reply:** Prein et al. (2023) assessed the model's skill in convection-permitting simulations of convection, precipitation, and snowfall over the TP by conducting multi-model, multi-physics ensemble simulations. One of the authors in this study was

involved in the study. According to analyses of precipitation simulations with different WRF physical parameterization schemes (PPSs), they found that there is little sensitivity to changing the model physics for simulations of precipitation patterns. Additionally, when using MYNN3 in combination with the Thompson scheme, the results are in much better performance. Thus, based on all results of precipitation simulations conducted by Prein et al. (2023), the PPSs used in our study are selected. This motivation will be added to the revised version.

Which albedo values are used in the LSM that the authors have chosen (page 6 line 158)? For example, in Noah-MP, the albedo parameterizations for land ice (variable ALBICE in phys/module_sf_noahmp_glacier.F) is set very high and might have to be changed to a value consistent with bare ice for more realistic simulations and atmospheric feedbacks. Please explain those values.

**Reply:** Considering the setting does not affect our results, we did not specifically modify the albedo parameterizations for land ice, and used the default settings of unified Noah LSM.

Minor comments:

Page 1 line 11: Please specify in the abstract what you mean with "different glacier and climate conditions" and add a sentence to better describe the methodology. Please also add some information in the abstract about the area and/or numbers of glaciers covered in the simulations.

**Reply:** Thanks for the comments. To evaluate the effects of glacier changes on precipitation under different climate backgrounds, we introduced two glacier conditions (i.e., the first and second Chinese glacier inventory) and two climate conditions (a warm year and a cold year) into WRF simulations. By comparing the outcomes of these simulations with different glacier and climate conditions, the relationship between glacier dynamics and precipitation responses can be elucidated. We will specify these mentioned above in the abstract, and add more descriptions of the methodology and glaciers covered in the simulations.

Page 1 line 19 and page 14 line 362: Can you please specify what you mean with "adequate"?

**Reply:** By "adequate" we mean that there is enough water vapor in the air to support precipitation.

Page 3 line 75-76: I suggest removing this sentence as it is a repetition.

**Reply:** Thanks for the suggestion, we will remove this sentence in the revision.

Page 4 Figure 1: I am unsure of what the 3 pink rectangles are. Please add an explanation in the caption, and add them to the legend. They are also very hard to see. Please use a different color and/or line width. Please also specify which data set you are using here for the elevation and cite it. Figure 1 should be referenced on page 4 line

**Reply:** Sorry for the confusion. The three pink rectangles are the locations of three typical regions selected for analyses in Section 4.3. We will add an explanation of the 3 pink rectangles in the caption, and add them to the legend in the revision. We will try to use a different color and/or line width to make these three rectangles look clearer. The elevation data used in Figure 1 are from the outcomes of the WRF model. Furthermore, we will add a reference to Figure 1 on page 4, line 90 of the text.

Page 4 line 99: Please specify what you mean with "fundamental parameters". Which parameters from this data set are you using for this study, in addition to the glacier outlines?

**Reply:** The fundamental parameters are area, volume, elevation, etc., which are also provided in the first and second Chinese glacier inventory. We will specify these mentioned above in the revision. In this study, only the glacier outlines are used.

Page 5 line 124: I think it would be helpful to add more information on the accuracy of the GPM data set based on previous studies.

**Reply:** Agreed. We will add more information on the accuracy of the GPM data set based on previous studies in the revision.

Page 5 line 131: Please introduce the WRF domain set-up and resolution before providing detailed information on the land cover change.

**Reply:** Point taken. We will reorganize the introduction of experimental design and model configuration.

Page 5 line 134: Please specify which land cover data you are using as input to WRF and at which resolution. Can you please also mention how exactly you changed the land cover – you are using TP locations based on shapefiles from Gl1 and Gl2. Which threshold for the intersection of the land cover and glacier shapefiles (i.e., Gl1 or Gl2 glacier percentage cover for the land cover grid cells; e.g. more than 50% of land cover grid cells covered by the glacier shapefiles) did you use when changing the categories? Additionally, can you please provide more details on the two different land use types (SNOW/ICE and bare ground) – what are differences in albedo and roughness length in the land use data set? Are there any other differences?

**Reply:** Land cover data used in this study was downloaded from the official website of the WRF model, which provides the geographical static data for WPS (WRF Preprocessing System) input. The resolution of land cover data is 30s, which is denoted by "usgs_lakes". The grid cells of glacier are identified by using the inpolygon function in MATLAB, which works based on the glacier shapefiles and locations of grid cells. The land use category of snow/ice is 24, and that of bare ground is 23. According to the file "LANDUSE.TBL", it can be seen that the albedo, surface moisture availability, surface emissivity, surface roughness, snow cover effect, thermal inertia constant, and surface heat capacity of two different land use types (snow/ice and bare ground) are different. The differences between snow/ice and bare ground in albedo and roughness

length in the land use data are as follows:

| Land Use | Season | Albedo | Surface Roughness |
|---|---|---|---|
| Snow/ice | Summer | 0.55 | 0.1 |
| | Winter | 0.70 | 0.1 |
| Bare ground | Summer | 0.25 | 10 |
| | Winter | 0.25 | 5 |

Page 6 line 150: "Grid spacing" and "resolution" refer to two different length scales and should not be used interchangeably (e.g., Grasso, 2000; Stull, 2015). Please use "grid spacing" here and for similar cases throughout the manuscript.

**Reply:** Point taken. We will correct the similar cases throughout the manuscript.

Page 6 line 151: Standard hourly WRF output is given as instantaneous variable values, and not hourly averages. Please clarify this in the text.

**Reply:** Point taken. We will clarify this in the revised manuscript.

Page 6 lines 153-158: Please include all references for the WRF physics options: https://www2.mmm.ucar.edu/wrf/users/physics/phys_references.html#LS

**Reply:** We will add all references for the WRF physics options.

Are you using the Unified Noah LSM? Please clarify and provide the correct reference.

**Reply:** Yes, we will clarify and provide the correct reference (Tewari et al., 2004) in the revision.

Reference:
Tewari, M., Chen, F., Wang, W., Dudhia, J., LeMone, M. A., Mitchell, K., et al. (2004). Implementation and verification of the unified NOAH land surface model in the WRF model. In 20th conference on weather analysis and forecasting/16th conference on numerical weather prediction (Vol. 1115). American Meteorological Society.

Page 6 Section 3.3.2: Please include the WRF time step and map projection.

**Reply:** The WRF time step and map projection are 60s and lambert, respectively, and we will add them to the revision.

Page 6 line 180: Are the authors representing and evaluating the inner WRF domain (D2) here (Fig. 2)? Please clarify.

**Reply:** Yes, we will clarify this in the revision.

Page 9 Figure 3: It is almost impossible to see the stippled lines in the plot. Please increase the line size. On page 8 line 10 the authors mention that "the relative differences in mean daily precipitation between the two glacier conditions are not statistically significant (p<0.1) for almost all grids". Can you please elaborate on how you define a whole region to have statistically significant differences?

**Reply:** We test the statistical significance of precipitation difference in each grid, not

for the whole region. What we want to describe here is that the relative differences in mean daily precipitation between the two glacier conditions are statistically significant (p<0.1) for only a few grid cells. Now we see that our description may cause a misunderstanding. Thus, we will modify the expression here, and increase the stippled lines in the plot. Additionally, we may explore whether the precipitation differences are statistically significant by looking at the results averaged over the whole region instead of the results for each grid to do the calculations.

Page 9 lines 220-222 and page 8 lines 199-201: Repetitions from Section 3.2.3. I suggest rephrasing in Section 3.2.3 (with regard to a reference data set).
**Reply:** Thanks for the comments, we will rephrase this section.

Page 9 line 228: Please provide some information on statistical significance here as well.
**Reply:** Thanks for the comment, we will add the statistical significance test here.

Figures 5 and 6: The thick red dots (lines) are overwriting the blue dots (lines). Please make sure to have both visible.
**Reply:** We will redraw figures 5 and 6 to make it look clearer.

Page 14 lines 314-319: Please explain in detail how you perturbed the initial conditions. Which variables were perturbed? By how much?
**Reply:** Referred to previous studies (Qiu et al., 2023), we perturb the initial conditions by reducing the spin-up time for the two additional simulations of warm year in summer by 1 hour and 3 hours, respectively. We will explain in detail the method of perturbing the initial conditions in the revision.

Reference:
Qiu, Y., Chen, J., Chen, D., Li, W. and Xiong, L. (2023) Lake-Area Expansion Alters Downwind Precipitation Patterns on the Tibetan Plateau: Insights From the Most Dramatically Expanded Lake. Journal of Geophysical Research: Atmospheres 128(15), e2023JD039274.

Page 14 lines 319-320: It is unclear which three regions the authors are talking about – from the text and the Figure 1. Please explain.
**Reply:** These three regions are selected based on the area and scale of glacier shrinkage. Thus, we labeled the locations of them in Figure 1. We will try to describe it in more detail in the revision.

Page 14 Section 4.3: Can you name Regions 1, 2, 3 based on, e.g., basins or subregions, instead of using 1, 2, 3?
**Reply:** Yes, we will consider renaming Regions 1, 2, and 3 in the revision.

Page 14 line 329: Analysis of precipitation changes vs. topographic height: Have you

analyzed possible correlations? The authors mention some correlations when discussing the water vapor flux for specific regions, but I am wondering whether the authors have conducted a more systematic analysis over the whole study area?

**Reply:** We have analyzed the relationship between precipitation changes and topographic height, but nothing special has been found. For the analyses of water vapor flux, we also have conducted over the whole study area. However, the whole study area is too large, so it is not suitable to analyze the meridional vertical profile of summer water vapor flux difference. As a result, we focused more on glacier changes.

Figures 7, 8, 9: It is very hard to see the pink dots for glacier-changed grid cells. Please increase the size of the dots. What do you mean by "glacier-changed" grid cells? Do you mean both disappeared and advanced glaciers (from Figure 1)? Please specify.

**Reply:** Yes, glacier-changed grid cells mean both disappeared and advanced glaciers. We will specify the mean by "glacier-changed" grid cells, and increase the size of dots in Figures 7, 8, and 9.

Page 18 line 404: Two typos

**Reply:** We will correct it.

Page 17 Discussion: Can you please explain in a few sentences the modeling of precipitation in the specific WRF physics schemes that you used in comparison to other WRF physics schemes, and possible implications for your results.

**Reply:** The modeling results of WRF physical parameterization schemes in different conditions may vary. It is hard to directly compare the specific WRF physical parameterization schemes used in this study with others in simulating precipitation.

Page 19 Conclusion: I believe it is important to mention statistical significance of the results here as well.

**Reply:** Thanks for your suggestion. We will add the statement of statistical significance of the results to the conclusions in the revision.

References:

Grasso, LD. (2000). The Differentiation between grid spacing and resolution and their application to numerical modeling. Bulletin of the American Meteorological Society. 81 (3). 579-580. 10.1175/1520-0477(2000)081<0579:CAA>2.3.CO;2.

Stull, R. B. (2015). Practical meteorology: An algebra-based survey of atmospheric science. Department of Earth, Ocean & Atmospheric Sciences, University of British Columbia, Vancouver, BC. https://doi.org/10.14288/1.0300441.